# Simulating Atmospheric Characteristics and Daytime Astronomical Seeing Using Weather Research and Forecasting Model

A. Y. Shikhovtsev [1,*], P. G. Kovadlo [1], A. A. Lezhenin [2], V. S. Gradov [3], P. O. Zaiko [4], M. A. Khitrykau [4], K. E. Kirichenko [1], M. B. Driga [1], A. V. Kiselev [1], I. V. Russkikh [1], V. A. Obolkin [5] and M. Yu. Shikhovtsev [5]

[1] Institute of Solar-Terrestrial Physics, The Siberian Branch of the Russian Academy of Sciences, Irkutsk 664033, Russia; kovadlo2006@rambler.ru (P.G.K.); kirichenko@isf.irk.ru (K.E.K.); driga@iszf.irk.ru (M.B.D.); kiselev@iszf.irk.ru (A.V.K.); vanekrus@iszf.irk.ru (I.V.R.)

[2] Institute of Computational Mathematics and Mathematical Geophysics, The Siberian Branch of the Russian Academy of Sciences, Novosibirsk 630090, Russia; lezhenin@ommfao.sscc.ru

[3] Departament of Mechanics and Mathematics, Novosibrsk State University, Novosibirsk 630090, Russia; v.gradov@g.nsu.ru

[4] Institute of Nature Management, National Academy of Sciences of Belarus, 220076 Minsk, Belarus; polly_lo@tut.by (P.O.Z.); m.a.hitrykau@gmail.com (M.A.K.)

[5] Limnological Institute, The Siberian Branch of the Russian Academy of Sciences, Irkutsk 664033, Russia; obolkin@lin.irk.ru (V.A.O.); max97irk@yandex.ru (M.Y.S.)

[*] Correspondence: ashikhovtsev@iszf.irk.ru; Tel.: +7-908-6464257

**Abstract:** The present study is aimed at the development of a novel empirical base for application to ground-based astronomical telescopes. A Weather Research and Forecasting (WRF) model is used for description of atmospheric flow structure with a high spatial resolution within the Baikal Astrophysical Observatory (BAO) region. Mesoscale vortex structures are found within the atmospheric boundary layer, which affect the quality of astronomical images. The results of simulations show that upward air motions in the lower atmosphere are suppressed both above the cold surface of Lake Baikal and inside mesoscale eddy structures. A model of the outer scale of turbulence for BAO is developed. In this work, we consider the seeing parameter that represents the full width at half-maximum of the point spread function. Optical turbulence profiles are obtained and daytime variations of seeing are estimated. Vertical profiles of optical turbulence are optimized taking into account data from direct optical observations of solar images.

**Keywords:** turbulence; mesoscale; WRF; telescope





## 1. Introduction

Astronomical observations in the optical, infrared and microwave bands of the electromagnetic spectrum are significantly limited by the Earth's atmosphere. The impact of atmosphere on the propagation of electromagnetic radiation is non-uniform in space and varies in time. The search for sites on the Earth with a minimal atmospheric effect on radiation is one of the most important astroclimatic tasks for the development of ground-based astronomical telescopes. In an astroclimate, depending on the light wavelength it is necessary to give preference to the study of various atmospheric characteristics [1–7]. In particular, optical observations require analysis of total cloud cover, scattered light, wind regime and, also, turbulent characteristics. This set of characteristics determines the possible observing time on an astronomical telescope and quality characteristics of astronomical images. In the infrared and microwave regions, statistical estimates of water vapor content and oxygen in different layers of the atmosphere are required. Water vapor and oxygen are the main atmospheric gases which determine the atmospheric optical thickness [8–11].

In astroclimatic studies, there are two problems. The first problem is related to the fact that measurements of atmospheric characteristics are often performed locally, at some given

sites, and cover not long enough time intervals of changes in the structure of atmospheric flows. The second problem is a consequence of the limited spatial and temporal resolution of atmospheric data. In order to increase the resolution, mesoscale models of the atmosphere and methods for parametrization of turbulent characteristics are being developed in the world. The Weather Research and Forecasting (WRF) model is widely used in simulations of atmospheric movements and pollution [12–17]. In recent years, the WRF model has been used to diagnose and predict optical turbulence as well as astronomical image quality characteristics (including seeing) [18–23].

This study is aimed at improvements in the methods for diagnosing atmospheric characteristics, which determine the quality of astronomical images and performances of adaptive optics systems [24].

## 2. Methods

In this paper, we analyze the possibility of simulation of the wind speed and temperature fields with a sufficiently high spatial and temporal resolution within the Baikal Astrophysical Observatory (BAO) region. The region includes three domains. The horizontal resolution in the internal domain is 500 m. The region is distinguished by a complex relief with strongly dissected mountains and plateaus.

In calculations, we used the non-hydrostatic WRF model [25]. This model is a powerful tool for the study of mesoscale phenomena in the atmosphere, precipitation and cloudiness. The basic equations of the WRF model are written as follows:

$$
\begin{cases}
\frac{\partial U}{\partial t} + m\left[\frac{\partial(Uu)}{\partial x} + \frac{\partial(Vu)}{\partial y}\right] + \frac{\partial(Wu)}{\partial \eta} + \left(\mu_d \alpha' \frac{\partial \overline{p}}{\partial x} + \mu_d \alpha \frac{\partial p'}{\partial x}\right) + \left(\frac{\alpha}{\alpha_d}\right)\left(\mu_d \frac{\partial H'}{\partial x} + \frac{\partial p'}{\partial \eta}\frac{\partial H}{\partial x} - \mu'_d \frac{\partial H}{\partial x}\right) = F_U, \\
\frac{\partial V}{\partial t} + m\left[\frac{\partial(Uv)}{\partial x} + \frac{\partial(Vv)}{\partial y}\right] + \frac{\partial(Wv)}{\partial \eta} + \left(\mu_d \alpha' \frac{\partial \overline{p}}{\partial y} + \mu_d \alpha \frac{\partial p'}{\partial y}\right) + \left(\frac{\alpha}{\alpha_d}\right)\left(\mu_d \frac{\partial H'}{\partial y} + \frac{\partial p'}{\partial \eta}\frac{\partial H}{\partial y} - \mu'_d \frac{\partial H}{\partial y}\right) = F_V, \\
\frac{\partial W}{\partial t} + m\left[\frac{\partial(Uw)}{\partial x} + \frac{\partial(Vw)}{\partial y}\right] + \frac{\partial(Ww)}{\partial \eta} - \frac{g}{m}\left(\frac{\alpha}{\alpha_d}\right)\left(\frac{\partial p'}{\partial \eta} - \overline{\mu_d}(q_{vapor} + q_{cloud} + q_{rain})\right) + \frac{g}{m}\mu'_d = F_W, \\
\frac{\partial H'}{\partial t} + \frac{1}{\mu_d}\left(m^2\left(U\frac{\partial H}{\partial x} + V\frac{\partial H}{\partial y}\right) + mW\frac{\partial H}{\partial \eta} - gW\right) = 0, \\
\frac{\partial \mu'_d}{\partial t} + m^2\left(\frac{\partial U}{\partial x} + \frac{\partial V}{\partial y}\right) + m\frac{\partial W}{\partial \eta} = 0, \\
\frac{\partial \theta}{\partial t} + m^2\left(\frac{\partial(U\theta)}{\partial x} + \frac{\partial(V\theta)}{\partial y}\right) + m\frac{\partial(W\theta)}{\partial \eta} = F_\theta, \\
\frac{\partial Q_m}{\partial t} + m^2\left(\frac{\partial(Uq_m)}{\partial x} + \frac{\partial(Vq_m)}{\partial y}\right) + m\frac{\partial(Wq_m)}{\partial \eta} = F_{Q_m}, \\
\frac{\partial \overline{H}}{\partial \eta} = -\overline{\mu}\cdot\overline{\alpha}, \\
\frac{\partial H'}{\partial \eta} = -\overline{\mu}\cdot\alpha' - \mu'\overline{\alpha}, \\
\eta = (p_h - p_{hupper})/\mu; \mu = p_{hsurface} - p_{hupper}
\end{cases}
$$

where $u$, $v$, $w$ are the wind speed components $U = \mu_d u/m$, $V = \mu_d v/m$, $W = \mu_d \dot{\eta}/m$, $\mu_d$ is the mass of dry air, $\dot{\eta}$ is the vertical velocity, $m$ is the scale coefficient (of the map). $\alpha = \alpha_d(1 + q_{vapor} + q_{cloud} + q_{rain})$ is the specific volume of dry air $\alpha_d$, water vapor, cloudiness and rain, $q$ is the ratio of mixture of water vapor, cloudiness and rain. The knowledge of these parameters within the region and at different heights makes it possible not only to estimate the vertical profiles of the structure constant of air refractive index fluctuations $C_n^2$, but also to determine the influence of mesoscale processes on the structured optical turbulence under different atmospheric conditions and the quality of astronomical images.

Interest in studies of mesoscale structures is supported by several factors. On the one hand, mesoscale structures arising in the atmosphere (mesoscale jets in the atmospheric boundary layer, mountain waves) distort the energy spectrum of turbulence and can enhance optical turbulence. On the other hand, mesoscale eddies can lead to the appearance of coherent turbulence and a decrease in turbulent phase and amplitude fluctuations.

In the simulations of atmospheric processes, we used a set of parametrization schemes presented in Table 1. These schemes determine the physics of atmospheric processes. In our opinion, the chosen parametrization schemes describe well the daytime structure of air flow and turbulence.

**Table 1.** Parametrization schemes in the WRF model.

| Physical Schemes of Parametrization | Description |
| --- | --- |
| Yonsei University scheme | Atmospheric boundary layer |
| MM5-similarity scheme | Surface layer |
| Kain–Fritsch scheme | Cloudiness |
| Simple scheme based on Dudhia | Short-wave radiation |
| Rapid Radiative Transfer Model scheme | Long-wave radiation |
| Thompson scheme | Microphysical processes in clouds |
| RUC scheme | Land surface model |

We must emphasize that, in simulation, we used an approach based on the turbulence coefficient $K_{turb}$. The parameter $K_{turb}$ is calculated using:

$$K_{turb} = k w_s z \left( 1 - \frac{z}{h_{ABL}} \right)^p,$$ (1)

$$w_s = u_* / \phi_m,$$ (2)

where $p = 2$, $k$ is the von Karman constant, $w_s$ is the mixed-layer velocity scale, $u_*$ is the friction velocity and $\phi_m$ is the non-dimensional function. This model is based on the parabolic $K$-profile in an unstable mixed layer with the addition of an explicit term to treat the entrainment layer at the top of the atmospheric boundary layer.

Moreover, we should note that both the atmospheric boundary layer and the surface layer are parametrized in the WRF model. Separate consideration of the surface layer is important as the most pronounced vertical gradients of wind speed, air temperature and, consequently, air refractive index are observed in this layer. A sufficiently high vertical resolution in the surface layer makes it possible to more accurately estimate the surface values of $C_n^2$.

In this study, we present the simulation results of the internal domain above the Baikal Astrophysical Observatory, with the highest horizontal resolution of 0.5 km, close to the scales of atmospheric small-scale turbulence. The time resolution is 3 min, with 44 vertical levels (from the ground to 30 km). The Baikal Astrophysical Observatory is located about 70 km from Irkutsk and the elevation is about 650 m above sea level. For the boundary and initial conditions, we make use of the Global Forecast System data.

The WRF model outputs the wind speed components, air temperature and other physical parameters at different heights above the ground. Figures 1–3 show typical daytime spatial distributions of wind speed at different heights in the atmosphere at the BAO site. The colorbar corresponds to atmospheric inhomogeneities with different wind speeds. The broken short arrows correspond to areas with atmospheric regions with low wind speeds. The long arrows correspond to regions with ordered structure of air flows. Knowing the curvature of air flows and the spatial scales of inhomogeneities, it is possible to judge atmospheric vorticity. The heights of the atmospheric layers are chosen taking into account the vertical scales of atmospheric inhomogeneities and the characteristic scales of turbulence. The heights correspond to the calculated positions of these layers. In addition, we use a higher vertical resolution within the lowest part of atmospheric boundary layer and near the tropopause. The variable vertical resolution is useful for correct estimation of the turbulent characteristics of atmospheric layers.

An analysis of the distributions shows that a mesoscale vortex structure in the atmospheric boundary layer is formed above BAO. This vortex structure develops in the lower layer of the atmosphere, up to heights ∼450–700 m. Dynamics of the vortex can be associated with the phenomenon of structured atmospheric turbulence. By modeling of such turbulence, the quality of the astronomical image can be significantly improved.

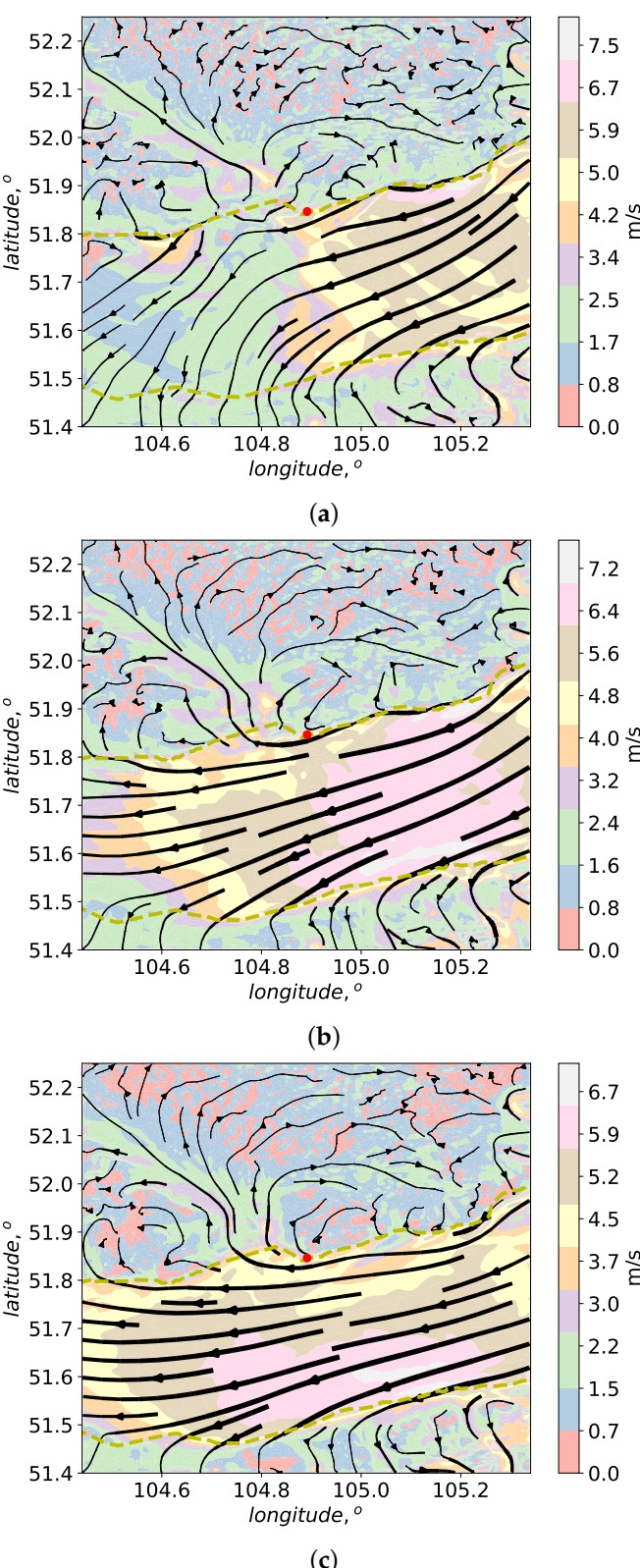

**Figure 1.** Characteristic daytime spatial distributions of wind speed derived from the WRF model at different heights in the atmosphere. The red marker shows the BAO site. The streamlines are indicated with arrows. The yellow dotted line is the coastline of Lake Baikal. (**a**) $z$ = 30 m. (**b**) $z$ = 101 m. (**c**) $z$ = 200 m.

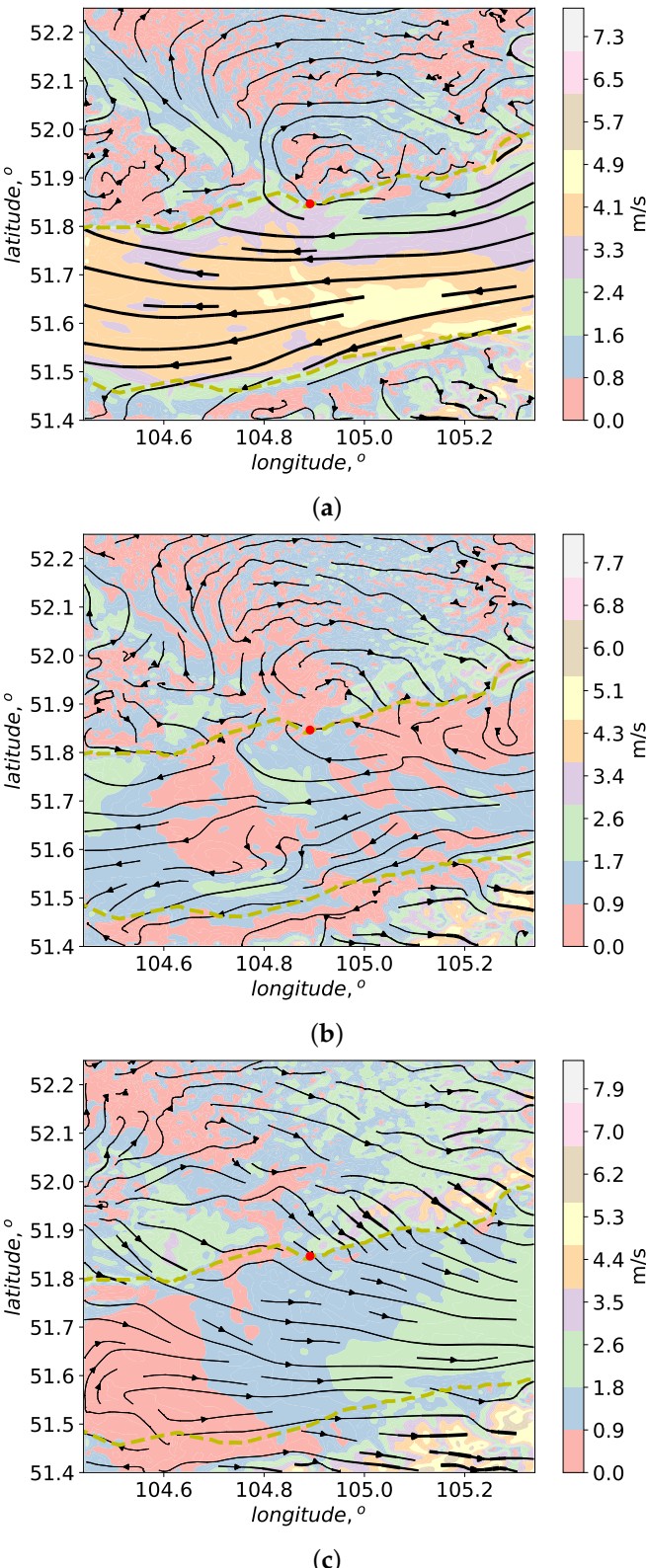

**Figure 2.** Continuation of Figure 1. (**a**) *z* = 326 m. (**b**) *z* = 485 m. (**c**) *z* = 687 m. The red marker shows the BAO site.

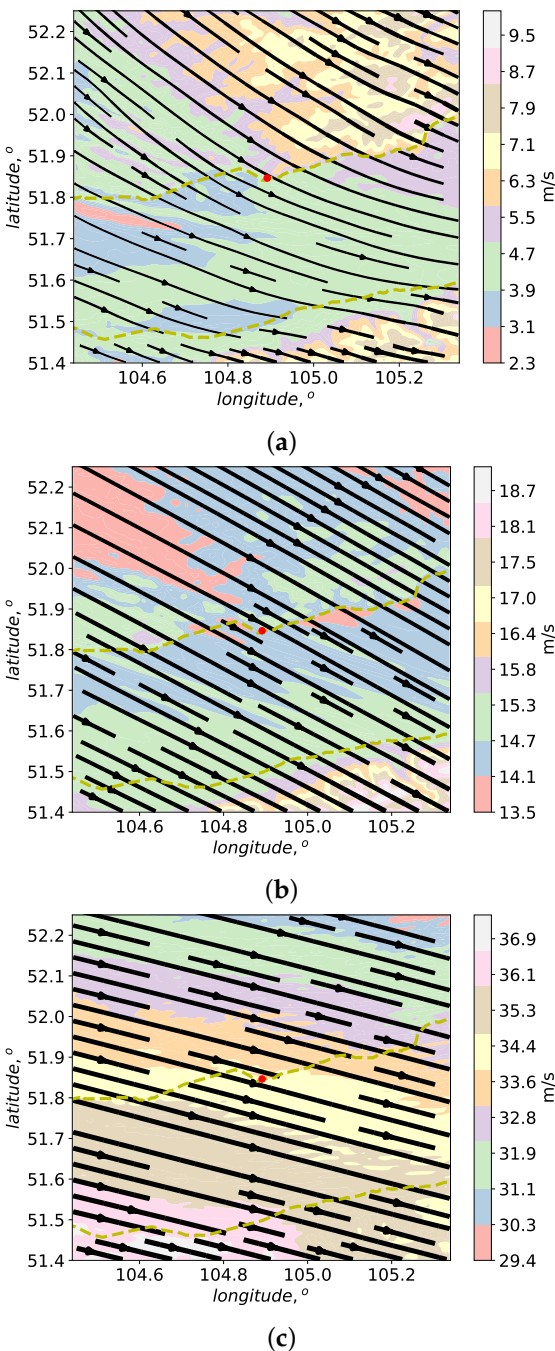

**Figure 3.** Continuation of Figure 1. (**a**) *z* = 1264 m. (**b**) *z* = 5555 m. (**c**) *z* = 12,148 m. The red marker shows the BAO site.

Another characteristic feature of the local air circulation is changing direction of wind at the heights from ∼450 m to ∼700 m above underlying surface (Figures 2 and 3). Against the background of a decrease in the mean wind speed in this layer, eddy movements also appear over Lake Baikal (mainly, at height ∼450–550 m).

Moreover, we may note that the structure of the air flows becomes more ordered with height. If the lower layers of the atmosphere are characterized by a high inhomogeneity of air flows, the atmospheric inhomogeneities in the upper layers are enlarged in size and stretched out by the mean airflow.

Figures 4–6 show the spatial distributions of calculated positive values of the vertical component of wind speed.

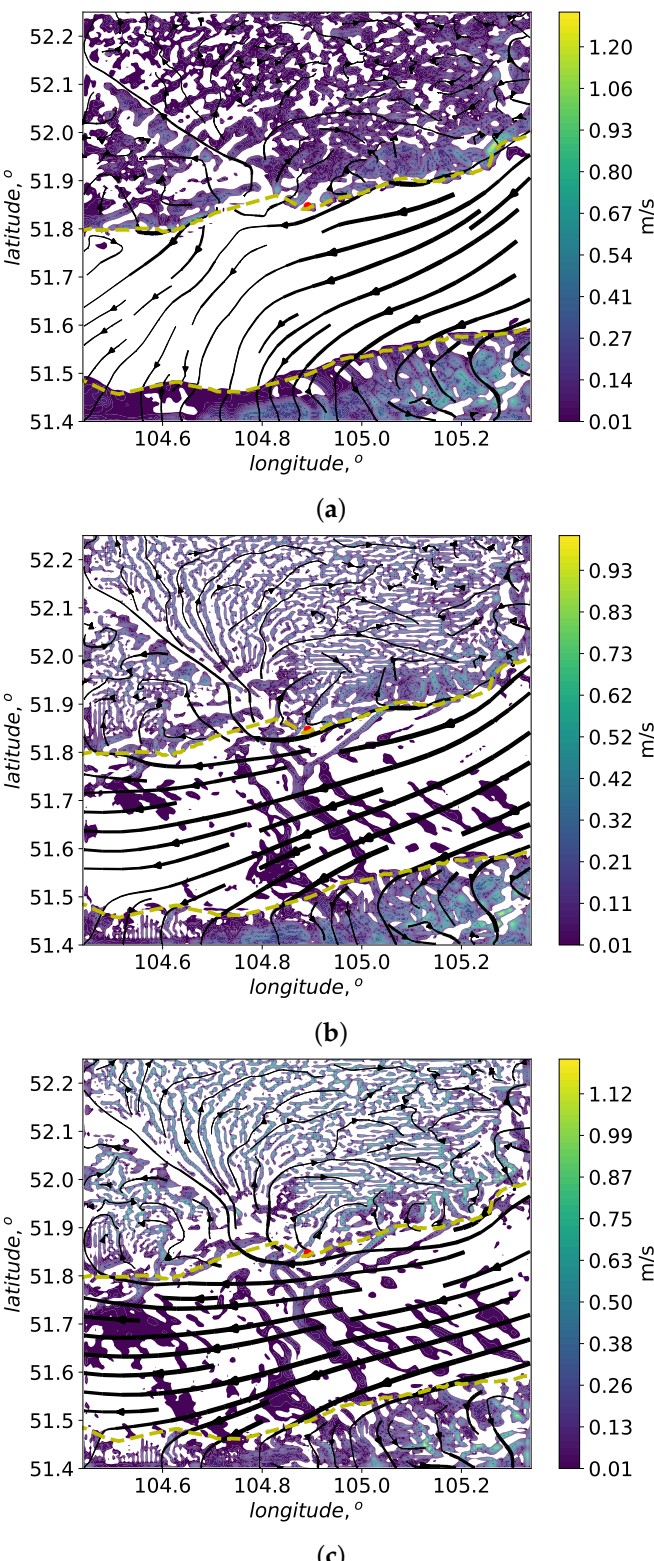

**Figure 4.** Characteristic daytime spatial distributions of positive vertical component of the wind speed at different heights in the atmosphere. The red marker shows the BAO site. The streamlines are shown with arrows. The yellow dotted line is the coastline of Lake Baikal. (**a**) $z = 30$ m. (**b**) $z = 101$ m. (**c**) $z = 200$ m.

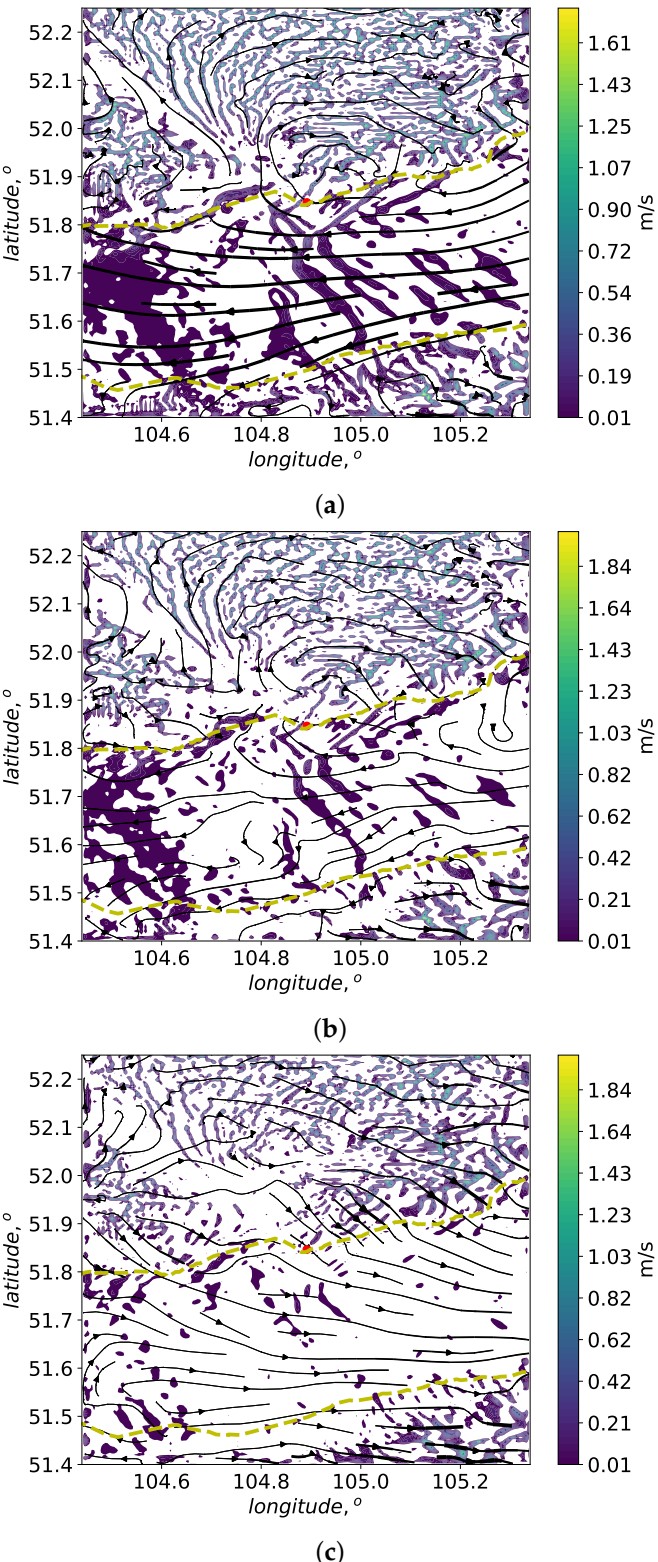

**Figure 5.** Continuation of Figure 4. (**a**) *z* = 326 m. (**b**) *z* = 485 m. (**c**) *z* = 687 m.

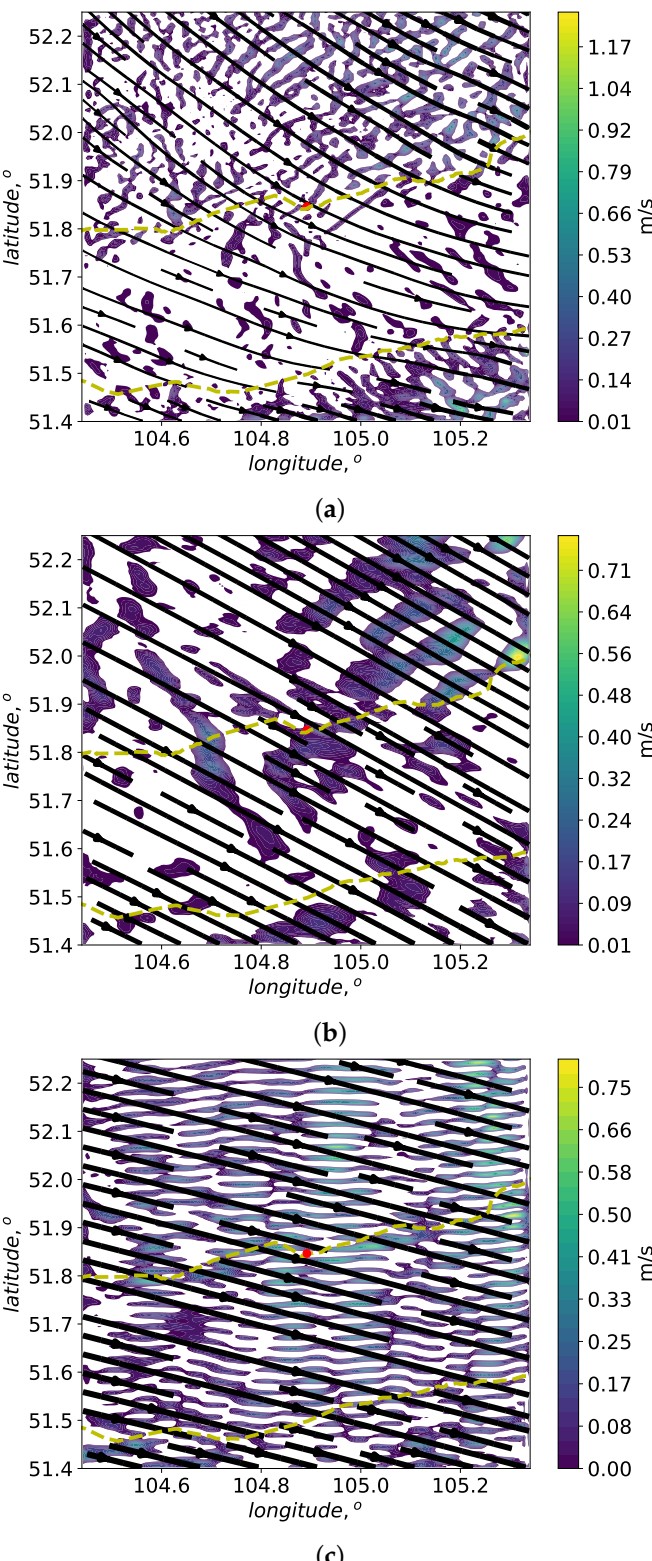

**Figure 6.** Continuation of Figure 4. (**a**) $z = 1264$ m. (**b**) $z = 5555$ m. (**c**) $z = 12{,}148$ m. The red marker shows the BAO site.

The figures show both the upward air movements and horizontal wind on the atmospheric level. The white areas represent negative values of the vertical component. The range of color changes is chosen for the convenience of comparing atmospheric characteristics at different levels in the atmosphere, as well as horizontal distributions of these

characteristics, within which local areas with strong upward movements are often observed. In the lower layers of the atmosphere above Lake Baikal, downward vertical movements are observed that suppress atmospheric turbulence. As the height increases, areas with positive vertical components of the wind speed are manifested. The spectral analysis of horizontal changes in the vertical wind speed shows that the characteristic linear scale of these atmospheric perturbations (with positive vertical movements) varies in the range from 3 to 8 km.

### 3. Calculated and Measured Seeing: WRF Model and Solar Image Motion Observations

In the paper, we estimate the structure constant of the air index refraction fluctuations $C_n^2$ along the line-of-sight of the Large Solar Vacuum Telescope (LSVT), which is the main astronomical instrument of the observatory (Figure 7). We should recall that a detailed characterization of the optical turbulence profile permits realistic modeling, performance prediction and real-time validation and optimization of adaptive optics instruments [26,27].

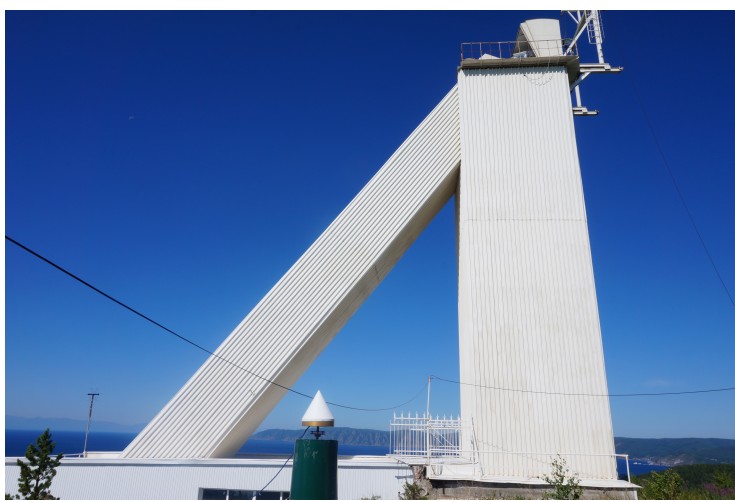

**Figure 7.** Large Solar Vacuum Telescope.

The integrated value of the constant allows us to estimate the image quality in terms of seeing. Atmospheric seeing represents the full width at half-maximum of the point spread function (PSF), the best angular resolution that an optical telescope can achieve in a long exposure image [28].

In order to verify the simulated characteristics, we compared the temporal changes of the seeing values derived from the WRF model and the measured values. Below, we present the main mathematical expressions used for calculation of the seeing parameter on the basis of WRF simulation data.

The seeing parameter associated with the full width at half-maximum of PSF may be estimated using the formula [29]:

$$seeing = \frac{0.98\lambda}{\left(0.423 sec\alpha \left(\frac{2\pi}{\lambda}\right)^2 \int_0^H C_n^2(z)dz\right)^{-3/5}},\tag{3}$$

where $H$ is the height of the optically active atmosphere, $\alpha$ is the zenith angle, $\lambda$ is the light wavelength ($\lambda$ = 500 nm). Parameter $C_n^2$ is related to meteorological characteristics of the atmosphere [30,31]:

$$C_n^2(z) = aL_0(z)^{4/3}M(z)^2,\tag{4}$$

where

$$M = \left(\frac{-79 \cdot 10^{-6}P}{T}\right)\frac{\partial ln\theta}{\partial z},\tag{5}$$

$$L_0^{4/3} = \begin{cases} 0.1^{4/3} \cdot 10^{1.64+42 \cdot S}, & \text{troposphere} \\ 0.1^{4/3} \cdot 10^{0.506+50 \cdot S}, & \text{stratosphere} \end{cases}, \tag{6}$$

$$S = \left[ \left( \frac{\partial u}{\partial z} \right)^2 + \left( \frac{\partial v}{\partial z} \right)^2 \right]^{0.5}, \tag{7}$$

where $L_0$ is the outer scale of turbulence, $z$ is the height, $P$ is the atmospheric pressure, $T$ is the air temperature, $\theta$ is the air potential temperature, $a$ is the constant, $u$ and $v$ are the horizontal components of wind speed $V(u, v, w)$. Thus, the vertical profiles of $C_n^2$ are calculated using the classical Dewan model, which takes into account the outer scale of turbulence and the vertical gradients of air refractive index $M$. Thus, the Dewan model is a convenient method for converting WRF-derived data into vertical profiles of $C_n^2$, the refractive index structure constant, which is the critical parameter for describing optical turbulence.

For the estimation of seeing, we also performed measurements of sunspot image motion at the Large Solar Vacuum Telescope (LSVT). The LSVT is a 760 mm telescope with elements of adaptive optics. Figure 8 shows a picture of the adaptive optics system of the Large Solar Vacuum Telescope. The optical layout of the system is simple. The system includes tip/tilt and deformable mirrors as well as a Shack–Hartmann wavefront sensor with different numbers of subapertures. In the Shack–Hartmann sensor, the incident wavefront is segmented by a microlens array and the formed spot displacements are measured. The measurements of image motion were taken using a Shack–Hartmann sensor sampled with six subapertures across the pupil. The main parameters of the Shack–Hartmann wavefront sensor are given in Table 2.

**Table 2.** Main parameters of Shack–Hartmann wavefront sensor.

| Parameter | Value |
| --- | --- |
| Aperture diameter | 760 mm |
| Used aperture diameter | 600 mm |
| Focal length | 40 m |
| Radiation wavelength | 535 nm |
| Number of subapertures | $6 \times 6$ |
| Equivalent size of subaperture | 10.0 cm |
| Angular pixel size | 0.3 ″/pix |
| Frame frequency | 100 Hz |
| Exposure | 30 ms |

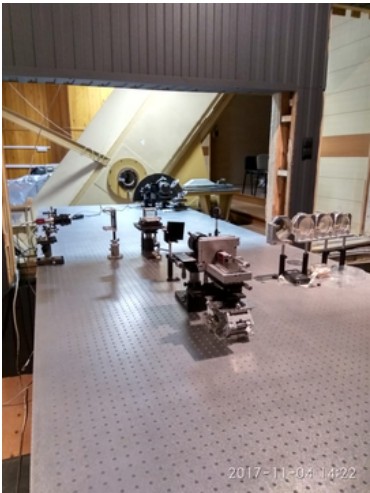

**Figure 8.** Adaptive optics system at the Large Solar Vacuum Telescope.

The subaperture size is 10.0 cm. The focal spot pattern detected via the Shack–Hartmann sensor is shown in Figure 9.

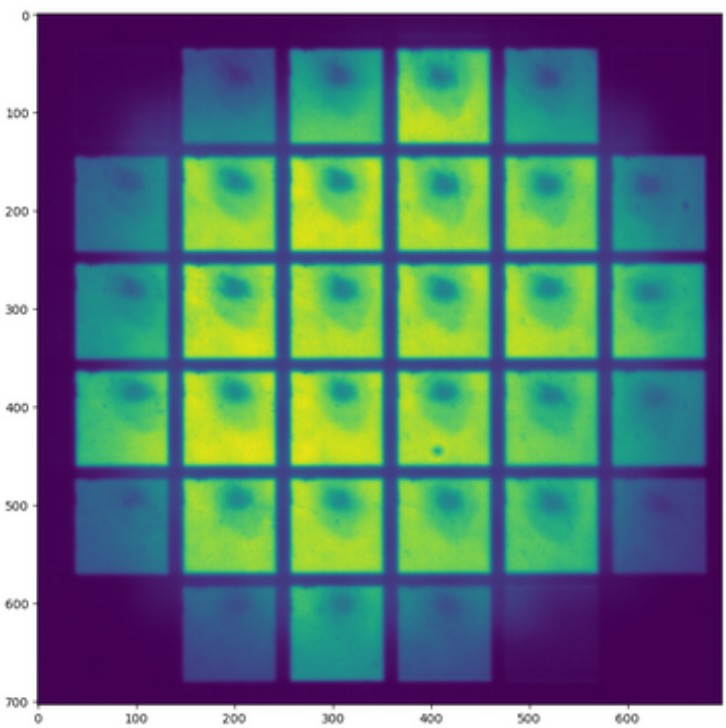

**Figure 9.** The focal solar spot pattern detected via the Shack–Hartmann sensor.

The system frame rate is 100 Hz. In measurements, we used light wavelength 535 nm. The seeing values from measurements were estimated using formulas (6)–(9) [29,32]:

$$seeing = \frac{0.98\lambda}{r_0}, \tag{8}$$

$$\sigma_\alpha^2 = K_i \lambda^2 r_0^{-5/3} D^{-1/3}, \tag{9}$$

where $\lambda$ is the light wavelength and $D$ is the telescope diameter. In formula (9), the Fried parameter $r_0$ is the quantity that characterizes the spatial scale of atmospheric turbulence, on which the dispersion of the phase fluctuations of a light wave is equal to 1 rad$^2$. The Fried parameter $r_0$ in formula (9) is a quantity estimated from the differential motion of solar images. The coefficient $K_i$ used in formula (9) depends on the ratio of the distance between the centers of the subapertures $S_d$ and the subaperture diameter $d_s$, as well as the direction of image motion and type of wavefront slope. In the present study, we used the formulas for longitudinal and transverse coefficients:

$$K_l = 0.34\left(1 - 0.57\left(\frac{S_d}{d_s}\right)^{-1/3} - 0.04\left(\frac{S_d}{d_s}\right)^{-7/3}\right), \tag{10}$$

$$K_t = 0.34\left(1 - 0.855\left(\frac{S_d}{d_s}\right)^{-1/3} + 0.03\left(\frac{S_d}{d_s}\right)^{-7/3}\right). \tag{11}$$

Figure 10 shows time changes in seeing measured with the help of a Shack–Hartmann sensor of the Large Solar Vacuum Telescope (8 August 2022).

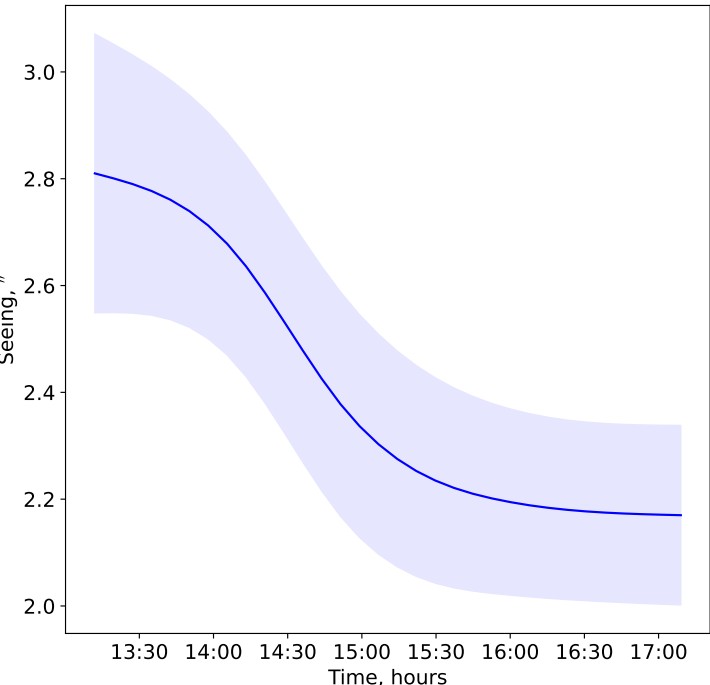

**Figure 10.** Seeing variations at the BAO site (8 August 2022). The blue line corresponds to changes in averaged seeing values. The fill shows the standard deviations of seeing.

Analysis of Figure 10 shows that, on average, the quality of the astronomical image improved in the evening: seeing values changed from 2.8″ to 2.2″. This behaviour, when seeing values decrease in the evening, is a classic scenario for the Baikal Astrophysical Observatory.

## 4. Modified Model for Estimation of $C_n^2$ Parameter at the Site of the Baikal Astrophysical Observatory

The structure of atmospheric turbulence changes significantly not only in time, but also depends on the site coordinates. In general, the turbulence is largely determined by the regional features of the interaction of airflows with the underlying surface as well as local factors of generation and dissipation of turbulence energy, as well as the transformation between the kinetic and potential energies of turbulence [33,34].

Following the classic Dewan model, the seeing values derived from WRF simulations are underestimated. For more accurate estimates, we analyzed the data of mast measurements of average and turbulent characteristics in the atmospheric surface layer at the BAO site. The measurements were taken using a meteorological ultrasonic complex. The measurements are based on the change in the time delay of ultrasonic waves in the air with different temperatures [35]. The complex is mounted on the Large Solar Vacuum Telescope near the primary mirror of the telescope at height 35 m above the ground, as shown in Figure 11. This complex is supplemented with the same sonic measurements of turbulent characteristics near the ground (at a height of 5 m). Thus, the anemometers make it possible to measure both turbulent characteristics, including fluctuations in the air refractive index at the height of the primary mirror and vertical gradients of meteorological characteristics in the atmospheric layer 5–35 m.

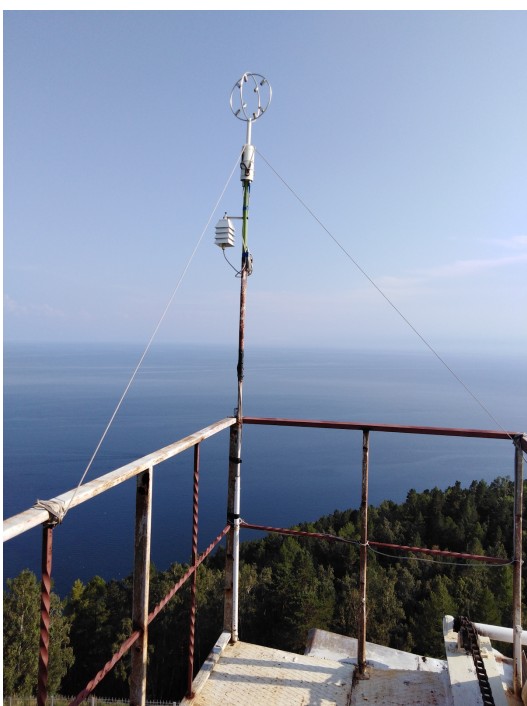

**Figure 11.** The meteorological ultrasonic complex mounted on the LSVT.

The atmospheric surface layer is a "laboratory" in which the relationships between the mean parameters and microturbulent characteristics of the atmosphere may be determined or clarified. The complex allows us to measure $C_n^2$ and vertical gradients of the horizontal components of wind speed. Using formulas (1)–(4), we compared the values of $C_n^2$ calculated on the basis of the weighted vertical wind speed gradients and the measured values of $C_n^2$. Figure 12 shows the temporal changes in the calculated and measured values of $C_n^2$. The calculated values of $C_n^2$ are obtained with the coefficients proposed by Dewan (Figure 12a). For convenience, we denote the coefficients as $a_{Ltsp}$ = 1.64 and $b_{Ltsp}$ = 42 s. Table 3 shows the statistics, including the mean absolute error ($MAE$), the standard deviation ($STD$) and the linear Pearson correlation coefficient ($K_{corr}$). As we can see, we obtain the Pearson correlation coefficient of 0.88, indicating a high degree of linear correlation between calculated and measured values of $C_n^2$ for coefficients $a_{Ltsp}$ = 2.5 and $b_{Ltsp}$ = 8 s.

The seeing values estimated from the WRF model are also shown in this table. The minimum values of $MAE$ and $STD$ between calculated and measured $C_n^2$ correspond to coefficients $a_{Ltsp}$ = 2.5 and $b_{Ltsp}$ = 8 s. The calculated values of $C_n^2$ are obtained with the coefficients proposed by Dewan (Figure 12a) and new coefficients (Figure 12b) selected by minimizing $MAE$ and $STD$.

**Table 3.** Statistical characteristics of models.

| $a_{Ltsp}$ | $b_{Ltsp}$ | $K_{cor}$ | MAE | STD | $L_{0median}$, m | Seeing, ″ |
|---|---|---|---|---|---|---|
| 42 | 1.64 | 0.48 | $1.1 \cdot 10^{-5}$ | $3.9 \cdot 10^{-4}$ | 13.0 | 1.15 |
| 2.1 | 10 | 0.80 | $1.1 \cdot 10^{-12}$ | $2.5 \cdot 10^{-11}$ | 7.7 | 2.25 |
| 2.5 | 8 | 0.88 | $8.0 \cdot 10^{-13}$ | $9.5 \cdot 10^{-12}$ | 6.0 | 2.43 |

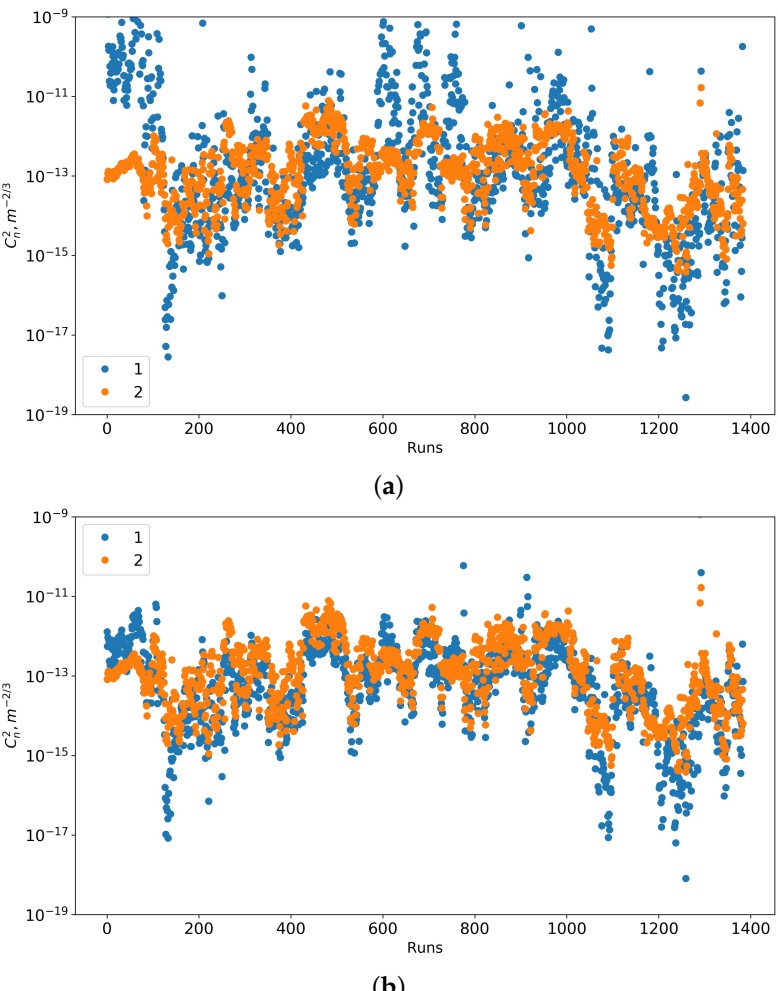

**Figure 12.** Temporary changes of calculated and measured values of $C_n^2$. The oranges markers correspond to the measured values of $C_n^2$. The blue markers correspond to the calculated values of $C_n^2$. (**a**) $a_{Ltsp}$ = 1.64, $b_{Ltsp}$ = 42 s. (**b**) $a_{Ltsp}$ = 2.5, $b_{Ltsp}$ = 8 s.

In the troposphere, the analysis of statistical characteristics shows that with an increase in $a_{Ltsp}$ and a decrease in $b_{Ltsp}$, the reproducibility of $C_n^2$ variations improves (the correlation coefficient increases; $MAE$ and $STD$ decrease). At the same time, we must emphasize that an increase in the coefficient $a_{Ltsp}$ leads to deformations of the vertical profiles of optical turbulence: optical turbulence strength in the upper atmosphere increases significantly.

Thus, the calculated profiles of $C_n^2(z)$ were obtained with new coefficients chosen by minimizing the mean absolute error and the standard deviation between the calculated and measured values $C_n^2$. We propose to use the following model to determine the vertical profiles of the outer scale of turbulence at the BAO site:

$$L_0^{4/3} = \begin{cases} 0.1^{4/3} \cdot 10^{2.5+8S}, Atmospheric\,Boundary\,Layer \\ 0.1^{4/3} \cdot 10^{1.64+42S}, Free\,Atmosphere \\ 0.1^{4/3} \cdot 10^{0.506+50S}, Stratosphere \end{cases}, \qquad (12)$$

Figure 13 demonstrates averaged daytime profiles of optical turbulence at the BAO site. The profile shown by the orange line is obtained from the well-known Hufnagel–Valley model, which is determined by the average surface value of $C_n^2$ and by pseudovelocity calculated for the atmospheric layer from 5 km to 20 km [36,37]. To estimate the surface values of $C_n^2$, we used data from mast measurements of turbulent characteristics (sonic

anemometers). Pseudovelocity was estimated according to Era-5 reanalysis data. The mean value of the pseudovelocity used in simulations was 23.7 m/s.

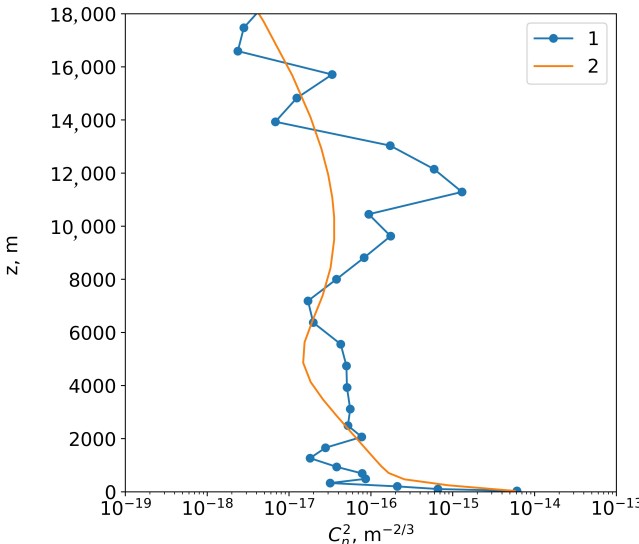

**Figure 13.** Average daytime profiles of optical turbulence at the BAO site. Line 1 corresponds to the profile calculated from WRF simulations. Line 2 corresponds to the Hufnagel–Valley model adapted for BAO.

Analyzing $C_n^2$ vertical profiles, we can note that the image quality is largely determined by the lower atmospheric layer and the turbulent layer at heights of 10–12 km. Compared to the Hufnagel–Valley model, the calculated profile $C_n^2$ is characterized by suppressed atmospheric turbulence in the lower layer up to a height of about 2 km. The WRF-derived profile $C_n^2$ also demonstrates a significant increase in the energy of optical turbulence at altitudes of 10–12 km. Estimating the $C_n^2$ profiles, we also evaluated the temporal changes in the seeing parameter during the day, shown in Figure 14. Analyzing this figure, we can note that the calculated and measured values of seeing are close.

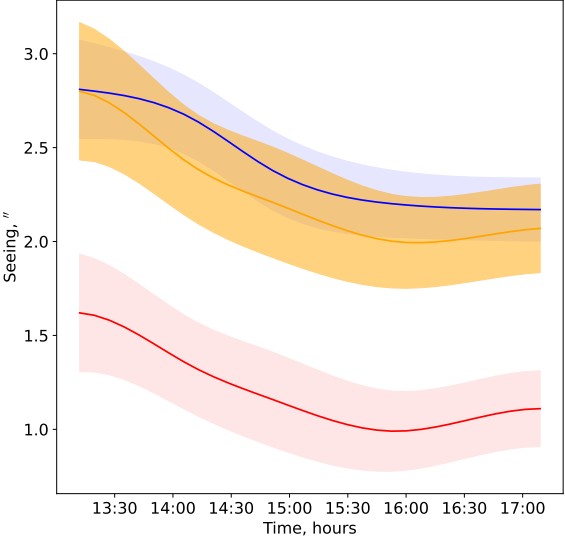

**Figure 14.** Temporary changes in the seeing parameter during the day. The blue line corresponds to the astronomical observations. The red line corresponds to the classic Dewan model. The orange line corresponds to the proposed model of vertical changes of the outer scale of turbulence.

### 5. Discussion and Results

Using the WRF model, vertical profiles of wind speed and air temperature over BAO were obtained under clear sky conditions. Based on the vertical profiles of these characteristics, we reconstructed the vertical profiles of the optical turbulence ($C_n^2$). It is shown that turbulence in the lower atmospheric layers is suppressed in comparison with the turbulence strength predicted via the Hufnagel–Valley model. A faster decrease in the optical turbulence strength with height is confirmed via experimental measurements of turbulent fluctuations over the cold water area of Lake Baikal under conditions of stable thermal stratification of the lower layers of the atmosphere. The correctness of the simulation results is, firstly, determined via the optimal parametrization schemes of atmospheric processes. Secondly, the simulation results are significantly affected by atmospheric conditions, which determine the characteristic ranges of changes in atmospheric characteristics, including the narrow limits of changes in the intensity of turbulent fluctuations and the turbulence coefficient under clear sky.

A model of the outer scale of turbulence for the Baikal Astrophysical Observatory is proposed. The coefficients found in the dependence of the outer scale on the vertical wind speed gradients at the site of BAO differ from the coefficients proposed by Dewan. We obtained the coefficients taking into account the minimization of the standard deviation and the mean absolute error between the measured and calculated values of $C_n^2$. We attribute these differences to the formation of mesoscale eddy structures above the observatory during the daytime. An analysis of vertical turbulence profiles shows that turbulence is suppressed in the lower atmosphere by mesoscale air movements that structure the atmospheric turbulence.

The seeing values derived from the WRF model are low in comparison with the measurement data. These low values are related to the fact that the WRF model underestimates the surface values of the structural constant of air refractive index fluctuations in comparison with the mast measurements.

**Author Contributions:** Investigation, visualization, writing—review and editing: A.Y.S. and P.G.K.; methodology: P.G.K., V.A.O., V.S.G., P.O.Z., M.A.K., A.Y.S. and A.A.L.; formal analysis, investigation, visualization: A.V.K., M.Y.S. and K.E.K.; writing—review and editing: A.Y.S., M.B.D. and P.G.K.; software, visualization: A.V.K., I.V.R. and M.Y.S. All authors have read and agreed to the published version of the manuscript.

**Funding:** This research was funded by the RSF grant No 22-29-01137.

**Institutional Review Board Statement:** Not applicable.

**Informed Consent Statement:** Not applicable.

**Data Availability Statement:** Data used are available on request from the corresponding author.

**Acknowledgments:** Measurements were carried out using the unique research facility large solar vacuum telescope, http://ckp-rf.ru/usu/200615/ (accessed on 1 March 2023).

**Conflicts of Interest:** The authors declare no conflict of interest.

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
