# Peer review of "Simulating Atmospheric Characteristics and Daytime Astronomical Seeing Using Weather Research and Forecasting Model"

_applsci, doi:10.3390/app13106354_

Round 1
Reviewer 1 Report
Some suggestions:
Title: The WRF model instead of Weather Research ...
P1-27: Did you mean limited number of sites?
"Short time intervals of changes" seems to be an advantage! Did you mean long data reporting time intervals in atmospheric stations?
P2-33, P2-45: The Weather Research and Forecasting (WRF) model or simply the WRF model. Not "A weather research and forecasting ..."
P4-fig1: Please explain the shaded and line contours separately, even both of them indicate the same variable.
P4-80: Are these 101m, 200m, 326m, etc, some random heights, or are they based on specific criteria? Why not considering round values of heights? such as 300, 500, 700? Why not based on pressure levels? Please indicate the Figure which is associated to your discussion. You could elaborate on the change in the wind speed. How was the directions? Etc.
P5-Fig2: Contrast of the background color with the wind streams is low.
Thickness of the lake's border could be increased.
P7-Fig4: Do this Figure shows only one variable of vertical wind speed, or two different variables of vertical wind speed and wind speed on atmospheric level? Please clearly mention it. If the white areas represent negative values, please clearly mention it in the caption. shaded contour plots must have such colors and values to easily distinguish the variations of the values through the domain. Here this is not the case.
P10-103: Reference of this equation?
P10-113: Please provide references
P3-76: With dynamics of the vortex ... can be replaced by "Dynamics of the vortex can be associated to the atmospheric turbulence"
P3-77: "With the development of ..." can be replaced by "By modeling of such turbulence".
Author Response
Dear we appreciate your dedicated time and effort in providing feedback on our manuscript. We are grateful for the insightful comments and valuable improvements to our paper. We have incorporated most of the suggestions made by you. Those changes are highlighted within the manuscript. Thank you very much.
Title: The WRF model instead of Weather Research …
The manuscript is written in application to astronomy. The WRF model is not widely known model among astronomers. In this regard, we ask you to agree with us and leave the full name of the model in the manuscript title and do not use the abbreviation.
P1-27: Did you mean limited number of sites?
"Short time intervals of changes" seems to be an advantage! Did you mean long data reporting time intervals in atmospheric stations?
In the manuscript we mean that measurements of atmospheric characteristics are performed at limited number of sites (However, it would be perfect to measure the atmospheric characteristics at a large number of sites. Moreover, even those measurements that are carried out are insufficient in duration)
P2-33, P2-45: The Weather Research and Forecasting (WRF) model or simply the WRF model. Not "A weather research and forecasting …"
We agree. We corrected it.
P4-fig1: Please explain the shaded and line contours separately, even both of them indicate the same variable.
We add information in the text.
P4-80: Are these 101m, 200m, 326m, etc, some random heights, or are they based on specific criteria? Why not considering round values of heights? such as 300, 500, 700? Why not based on pressure levels? Please indicate the Figure which is associated to your discussion. You could elaborate on the change in the wind speed. How was the directions? Etc.
We add information in the text.
P5-Fig2: Contrast of the background color with the wind streams is low.
Thickness of the lake's border could be increased.
We re-plotted the figures
P7-Fig4: Do this Figure shows only one variable of vertical wind speed, or two different variables of vertical wind speed and wind speed on atmospheric level? Please clearly mention it. If the white areas represent negative values, please clearly mention it in the caption. shaded contour plots must have such colors and values to easily distinguish the variations of the values through the domain. Here this is not the case.
We add information in the text. From the point of view of the development of turbulence, we are interested only in those areas in the atmosphere that are characterized by increased velocities of upward movements.
P10-103: Reference of this equation?
We agree. We add reference.
P10-113: Please provide references
We agree. We add reference.
Comments on the Quality of English Language
P3-76: With dynamics of the vortex ... can be replaced by "Dynamics of the vortex can be associated to the atmospheric turbulence"
P3-77: "With the development of ..." can be replaced by "By modeling of such turbulence".
We agree. We corrected the sentences

Reviewer 2 Report
Review Report is attached.

Author Response
Review Report: In this manuscript, authors study the empirical improvements in the atmospheric characteristics for the ground-based telescopes. They use a Weather Research and Forecasting (WRF) model to characterize these effects, with a high spatial resolution, within the Baikal Astrophysical Observatory. In my opinion, this manuscript deserves publication after addressing a few queries.
Dear reviewer we highly appreciate your dedicated time and effort in providing feedback on our manuscript. We are grateful for the insightful comments and valuable improvements to our paper. We try to take into account most of the excellent suggestions made by you. Those changes are highlighted within the manuscript.
My comments are the following:
General:
1. In Section 2, a precise description of the WRF model with some physics arguments is needed for better understanding of the work to the readers.
We try to add some information in the manuscript, in particular, about the daytime turbulence of the atmospheric boundary layer (this layer generates the phase fluctuations of light waves (mainly) and, thus, distortions of astronomical images. The detailed description of the model is very large in volume, the model contains many parameterization schemes and their combinations, and is also based on the solution of hydrothermodynamic equations.
2. Authors only discuss the day-time characteristics of the atmosphere. It will be interesting to comment on the night-time characteristics as optical and IR telescopes are operated during night times.
The characteristics of the atmospheric boundary layer are significantly different during day-time and night time. The turbulence structure also differs and, as a consequence, the parameterization scheme and physical parameters (including vertical changes in the turbulence coefficient) should be changed. We compared the simulation data with the observational data from the solar telescope. The study of night-time turbulence is of independent interest (future study). The complexity of joint modeling (night-day) is due to the fact that at night, the thickness of the boundary layer, as well as the intensity of turbulence, decreases significantly. The physical dependences between turbulent and mean characteristics also change.
Minor:
Abstract
Line 2: Weather Research And Forecasting Model (WRF) is => Weather Research and Forecasting (WRF) Model is
We agree. We corrected.
Line 5: results of simulations shows that => results of simulations show that
We agree. We corrected.
Line 8: In the paper, we consider parameter seeing that represents => In this work, we consider the seeing parameter that represents
We agree. We corrected.
Introduction
Line 14: microwave ranges of the => microwave bands of the
We agree. We corrected.
Line 16: atmosphere on propagation => atmosphere on the propagation
Line 24: infrared and microwave ranges => infrared and microwave regions
Line 34: the WRF model was used to => the WRF model has been used to
Line 37: improvements of the methods => improvements in the methods
Methods Line 45: non-hydrostatic Weather Research and Forecasting (WRF) model => non-hydrostatic WRF model Line 66: In the study we present => In this study, we present
We agree. We corrected.
Line 103: Please provide appropriate reference
We add reference
Line 112-113: Cite a reference for the Dewan Model. Also, explaining this model briefly.
We cited the paper and add some information about Dewan model.
Line 123: sensor is shown => sensor are given
We agree. We corrected.
Line 127: Please provide appropriate reference for these formula. Also, briefly discuss the difference between Equation (1) and (6) relating to the seeing parameter.
We add reference and add some information.
Line 131: coefficient KI => coefficient Ki
We agree. We corrected.
Line 170: Briefly explain, which correlation Kcorr stands for ? And how it has been computed ?
We add some information. We used classical coefficient, namely, the linear Pearson correlation coefficient.
Line 175: analysis of statistical characteristics show that => analysis of statistical characteristics shows that
We agree. We corrected.
Line 185: Cite a reference for Hufnagel-Valley model.
We add reference
Discussion
Line 221: structure atmospheric turbulence => structure the atmospheric turbulence.
We add reference
Line 222: WRF => WRF model
We add reference

Reviewer 3 Report
The article reported the experimental results with the WRF model and represented the different viewpoints for the seeing of the optical turbulence. It will serve as a meaningful reference for the design and application of telescope systems. I suggest accept the article.
Please note that there is Baikal in Line 66 instead of Baykal.
Author Response
Dear reviewer we highly appreciate your dedicated time and effort in providing feedback on our manuscript. We are grateful for your attention to us. Thank you.
